# Integrated Effects of Soil Moisture on Wheat Hydraulic Properties and Stomatal Regulation

**DOI:** 10.3390/plants13162263

**Published:** 2024-08-14

**Authors:** Lijuan Wang, Yanqun Zhang, Dandan Luo, Xinlong Hu, Pancen Feng, Yan Mo, Hao Li, Shihong Gong

**Affiliations:** 1State Key Laboratory of Simulation and Regulation of Water Cycle in River Basin, China Institute of Water Resources and Hydropower Research, Beijing 100038, China; 2Department of Irrigation and Drainage, China Institute of Water Resources and Hydropower Research, Beijing 100038, China; 3School of Geography and Planning, Jining Normal University, Jining 012000, China

**Keywords:** winter wheat, soil water stress, leaf water potential, leaf hydraulic conductance, stomatal conductance, stomatal trait

## Abstract

The development of water-saving management relies on understanding the physiological response of crops to soil drought. The coordinated regulation of hydraulics and stomatal conductance in plant water relations has steadily received attention. However, research focusing on grain crops, such as winter wheat, remains limited. In this study, three soil water supply treatments, including high (H), moderate (M), and low (L) soil water contents, were conducted with potted winter wheat. Leaf water potential (*Ψ*_leaf_), leaf hydraulic conductance (*K*_leaf_), and stomatal conductance (*g*_s_), as well as leaf biochemical parameters and stomatal traits were measured. Results showed that, compared to H, predawn leaf water potential (*Ψ*_PD_) significantly reduced by 48.10% and 47.91%, midday leaf water potential (*Ψ*_MD_) reduced by 40.71% and 43.20%, *K*_leaf_ reduced by 64.80% and 65.61%, and *g*_s_ reduced by 21.20% and 43.41%, respectively, under M and L conditions. Although *g*_s_ showed a significant difference between M and L, *Ψ*_leaf_ and *K*_leaf_ did not show significant differences between these treatments. The maximum carboxylation rate (*V*_cmax_) and maximum electron transfer rate (*J*_max_) under L significantly decreased by 23.11% and 28.10%, stomatal density (SD) and stomatal pore area index (SPI) under L on the abaxial side increased by 59.80% and 52.30%, respectively, compared to H. The leaf water potential at 50% hydraulic conduction loss (P_50_) under L was not significantly reduced. The *g*_s_ was positively correlated with *Ψ*_MD_ and *K*_leaf_, but it was negatively correlated with abscisic acid (ABA) and SD. A threshold relationship between *g*_s_ and *K*_leaf_ was observed, with rapid and linear reduction in *g*_s_ occurring only when *K*_leaf_ fell below 8.70 mmol m^−2^ s^−1^ MPa^−1^. Our findings demonstrate that wheat leaves adapt stomatal regulation strategies from anisohydric to isohydric in response to reduced soil water content. These results enrich the theory of trade-offs between the carbon assimilation and hydraulic safety in crops and also provide a theoretical basis for water management practices based on stomatal regulation strategies under varying soil water conditions.

## 1. Introduction

Soil drought stress is a critical constraint to grain crop production, and the increasing frequency and intensity of soil drought are exacerbating crop water demand [1]. To cope with this challenge, plants have evolved a great number of physiological adaptive mechanisms, especially for the regulation of stomatal conductance (*g*_s_) as a water-saving mechanism, which has steadily received attention [2]. However, research focusing on grain crops, such as winter wheat, is still insufficient, including the coordinated regulation of hydraulics and stomatal conductance in plant water relations. Therefore, it is necessary to explore the complex relationship between stomatal behavior, leaf water status, and morphological adaptations of stomata in response to various drought stresses. This is essential for optimizing water use efficiency in agricultural production.

There have been several studies on the effects of soil drought on hydraulic and stomatal indices. For example, Wang et al. [3] demonstrated that soil water stress significantly decreased *g*_s_ in winter wheat, highlighting the plant’s vulnerability to drought conditions. The midday leaf water potential (*Ψ*_MD_) in field winter wheat [4] and potted rice [5] decreased significantly under soil water stress. Soil water stress was shown to diminish leaf hydraulic conductance (*K*_leaf_) in *Arabidopsis* and olive, as documented by Hernandez et al. [6] and Scoffoni et al. [7]. Previous studies showed that the variable *K*_leaf_ was closely related to the response of *g*_s_ to drought stress in two Mediterranean fruit tree species [6]. The sensitivity of the plant to drought can be assessed by fitting a vulnerability curve to the relationship between *K*_leaf_ and *Ψ*_leaf_. Xylem water potential at 50% hydraulic conduction loss (P_50_) can be derived from this curve, serving as an indicator of the plant’s susceptibility to drought. A higher P_50_ value indicates greater vulnerability of the crop to drought [8]. Stiller et al. [9] identified xylem cavitation as the primary factor in the reduction of the *K*_leaf_ of rice during soil drought. These findings provide valuable insights into the effects of soil drought on key hydraulic and stomatal indices, but it is necessary to consider the integrated effects of soil moisture on plant hydraulic properties and stomatal regulation.

Based on the sensitivity of stomata to hydraulic imbalance induced by drought, strategies for regulating stomata are categorized into isohydric and anisohydric types [10]. There is an ongoing debate among researchers about whether *K*_leaf_ is the primary factor behind stomatal closure. Studies have shown that the decrease in *g*_s_ was not primarily influenced by *K*_leaf_. For instance, Corso et al. [11] proposed that during varying degrees of soil water stress, the decline in *K*_leaf_ did not predominantly control stomatal closure in wheat under soil water stress conditions ranging from 0 to −5 MPa. On the contrary, some studies suggested that *K*_leaf_ was the main driving force for the decrease in *g*_s_. For instance, Tombesi et al. [12] discovered that in grapes, stomatal closure due to drought was initially caused by a drop in *K*_leaf_ and was maintained by abscisic acid (ABA). Wang et al. [5] reported that during dehydration, *K*_leaf_ was the primary driver of stomatal closure and reduction in *g*_s_ under soil drought stress in rice. In addition, the decrease in *g*_s_ could be attributed to other factors besides *K*_leaf_. For instance, Aasamaa et al. [13] found that the reduction in *g*_s_ was predominantly governed by *Ψ*_leaf_ under drought conditions in tomatoes. Abdalla et al. [14] identified that the reduction in root hydraulic conductivity was the primary driver of stomatal closure in tomato under soil drought. Hence, it is necessary to further investigate the relationship between stomatal behavior and leaf water status (e.g., *Ψ*_leaf_ and *K*_leaf_) in winter wheat under soil drought stress.

Crops can adapt to drought-induced environmental changes through both short-term control of stomatal pore size and long-term control of stomatal development and morphology [15]. Stomatal traits are important factors that affect *g*_s_ [16]. Numerous studies have focused on the response of stomatal length (SL_en_) and stomatal density (SD) under soil drought stress. The stomata traits of different crops, or even the same crops, reflect different responses to drought stress. For instance, Li et al. [17] observed that wheat exposed to drought exhibited reduced SL_en_ on both the adaxial and abaxial sides, accompanied by a negative correlation between SD and *g*_s_. Zhao et al. [18] reported that drought notably increased SD and caused a reduction in both stomatal pore sizes in maize. Studies on *Leymus chinensis* showed that moderate soil drought increased SD, while severe drought decreased SD. Xu et al. [19] and Yang et al. [20] discovered that drought induced a gradual increase in adaxial SD in tomato, while adaxial SD initially decreased and then increased. Although the present study has revealed the variability of stomatal responses to drought stress across different species and even within the same species, it remains unclear how these stomatal traits interact with other physiological indices under different intensities and durations of soil drought stress. 

At present, there are few studies on the stomatal traits of wheat under various soil drought stress conditions and their quantitative relationships with physiological parameters. Previous studies have primarily focused on the response of crops to water stress in terms of physiological and growth parameters. However, there has been less emphasis on understanding the internal mechanisms of *g*_s_ in response to drought stress through leaf water status and stomatal traits. In this study, winter wheat was subjected to high, moderate, and low soil water supply treatments. Various parameters including the *g*_s_, *Ψ*_leaf_, *K*_leaf_, maximum carboxylation rate (*V*_cmax_), maximum electron transfer rate (*J*_max_), stomatal morphology traits, relative water content (RWC), and ABA of leaves were measured. The aims of this study were to analyze how these parameters respond to different soil drought stress and how they interrelate, thus revealing the physiological mechanism of winter wheat’s drought tolerance under various soil water conditions. It is of great scientific significance to explain the internal causes of *g*_s_ changes caused by different drought stress treatments in terms of plant physiology and stomatal characteristics in order to enhance wheat water use efficiency.

## 2. Results

### 2.1. Stomatal Conductance, Leaf Hydraulic Conductivity and Leaf Water Potential Response

There was no significant difference in *K*_leaf_ between the M and L soil water supply. However, compared to H, *K*_leaf_ significantly reduced by 64.80% in M and 65.61% in L (Figure 1a). Specifically, *g*_s_ significantly reduced by 21.20% in M and 43.41% in L, compared to H (Figure 1b). *Ψ*_PD_ and *Ψ*_MD_ did not significantly change between M and L. However, when compared to H, *Ψ*_PD_ significantly reduced by 48.10% in M and 47.91% in L, and *Ψ*_MD_ significantly reduced by 40.71% in M and 43.20% in L (Figure 1c). 

The relationship between *g*_s_ and *K*_leaf_ was found to be piecewise linear (R^2^ = 0.49, *p* = 0.049). When *K*_leaf_ exceeded 8.70 mmol m^−2^ s^−1^ MPa^−1^, *g*_s_ decreased at a slower rate. Conversely, when *K*_leaf_ was less than 8.70 mmol m^−2^ s^−1^ MPa^−1^, *g*_s_ decreased more rapidly (Figure 2a). Similarly, a piecewise linear relationship was observed between *g*_s_ and *Ψ*_MD_ (R^2^ = 0.67, *p* = 0.004). When *Ψ*_MD_ was above −1.48 MPa, *g*_s_ decreased slowly; however, when *Ψ*_MD_ dropped below −1.48 MPa, *g*_s_ decreased rapidly (Figure 2b). *K*_leaf_ and *Ψ*_MD_ presented a significant positive correlation (R^2^ = 0.46, *p* = 0.004) (Figure 2c).

The P_50_ was estimated as −1.04 and −1.01 MPa in H and M, respectively. The slope (S) of the vulnerability curve at the inflection point (P_50_) was estimated as −58.3% and −52.67% in H and M, respectively. The overlapping confidence intervals for the hydraulic vulnerability curves of the leaves suggest that, when assessed at a 95% confidence level, there was no statistically significant difference between H and M treatments (Figure 3).

### 2.2. Biochemical Indices of Leaves

The *V*_cmax_ and *J*_max_ varied from 65.73 to 89.87 µmol m^−2^ s^−1^ and 117.14 to 223.22 µmol m^−2^ s^−1^, respectively. The *V*_cmax_ and *J*_max_ of L significantly reduced by 23.11% and 28.10%, respectively, compared to H. There was no significant difference in *V*_cmax_ and *J*_max_ between M and H (Table 1). The RWC of the L treatment significantly reduced by 11.47% compared to H. There was no significant difference in RWC between H and M. ABA significantly increased by 69.35% in M and 89.32% in L.

There was a significant positive correlation between *g*_s_ and *V*_cmax_ (R^2^ = 0.64, *p* < 0.001), and *g*_s_ and *J*_max_ (R^2^ = 0.35, *p* = 0.019) (Figure 4a,b). A significant negative correlation was found between *g*_s_ and ABA (R^2^ = 0.81, *p* < 0.001) (Figure 4c).

### 2.3. Stomatal Traits

There were no significant differences in SL_en_ on both the adaxial and abaxial leaf sides among three soil water levels (Figure 5a). The SD on the adaxial side significantly increased by 81.82% in L compared to H. The SD on the abaxial side significantly increased by 24.71% in M and 59.80% in L compared to H (Figure 5b). The stomatal pore index (SPI) on the abaxial side significantly increased by 52.30% in L compared to H. Conversely, there was no significant difference for SPI on the adaxial side among the three soil water levels (Figure 5c).

A strong negative correlation was observed between *g*_s_ and SD on the adaxial side (R^2^ = 0.73) and on the abaxial side (R^2^ = 0.74). In contrast, a significant correlation between *V*_cmax_ and SD on the adaxial side (R^2^ = 0.53) and on the abaxial side (R^2^ = 0.52) was found in Figure 6. 

## 3. Discussion

It is crucial to understand how plants respond to drought stress, particularly given the ongoing global climate change. The responses of crops to drought are primarily shaped by imbalances in their water transport systems and the regulation of stomatal movements. Our findings further support this view and reveal how crops adapt to drought conditions by balancing *g*_s_ and *K*_leaf_. As previously documented [21], crops absorb CO_2_ through their stomata while releasing water vapor, demonstrating the intricate relationship between *g*_s_ and *K*_leaf_. 

Our study revealed significant physiological adaptations in wheat plants under varying soil drought stress. As soil water content declined to 65–75% of field capacity (FC), plants under M showed significantly lower *Ψ*_PD_ and *Ψ*_MD_, *K*_leaf_, and *g*_s_ compared to H. Notably, the degree of reduction in *Ψ*_PD_ and *Ψ*_MD_ was approximately one-third of the *K*_leaf_ reduction, while the reduction in *g*_s_ was nearly two-thirds of the *K*_leaf_ reduction. This indicated that in early soil drought stress, plants prioritize limiting water loss by reducing *g*_s_, followed by a decrease in *K*_leaf_. Meanwhile, photosynthetic capacity parameters (i.e., *V*_cmax_ and *J*_max_) remained stable, suggesting that photosynthesis is initially preserved. However, as drought intensified to 45–55% FC, *g*_s_, *V*_cmax_, and *J*_max_ declined significantly, indicating a sequential physiological response. The *K*_leaf_ and *Ψ*_leaf_ did not continue to decrease significantly with increasing soil drought intensity, while *g*_s_ showed consistent reductions, highlighting its sensitivity to soil drought stress. These findings demonstrated a shift in wheat strategies from favoring growth to prioritizing survival under soil drought, reflecting the complexity of plant adaptations to soil drought stress.

The results of our study confirm previous findings that wheat plants undergo a significant decline in leaf water status under soil drought stress. Specifically, Nayyar et al. [22] reported that RWC in wheat leaves decreased by 34.8% when soil moisture dropped by 46.7%. These findings are consistent with our observations, where RWC significantly declined by 11.5% in L compared with H. Akter et al. [4] found a 39.5% reduction in the *Ψ*_PD_ of wheat under conditions when soil water content is around 18%. We also observed that both *Ψ*_PD_ and *Ψ*_MD_ under M and L similarly reduced compared to H (Figure 1c). Previous studies have demonstrated that under drought stress, the reduction in *K*_leaf_ directly impacts both *Ψ*_leaf_ and stomatal regulation [23]. In our study, we also observed that *g*_s_ showed a significant positive correlation with *Ψ*_leaf_ (Figure 2b). This finding was consistent with previous research [24], which demonstrated a strong positive correlation between the decrease in *Ψ*_leaf_ and *g*_s_ during soil drought in tomato.

There have been few studies focusing on the driving factors that influence stomatal behavior under soil water stress. Müllers et al. [23] reported the impact of soil water stress on *Ψ*_leaf_ and noted the role of *Ψ*_leaf_ in facilitating stomatal closure in maize. This observation was consistent with our own study, as illustrated in Figure 2b, where similar patterns of stomatal response to water stress were observed. The difference in stomatal behavior was observed in our study when *K*_leaf_ dropped below 8.70 mmol m^−2^ s^−1^ MPa^−1^. As *K*_leaf_ decreased, *g*_s_ reduction rapidly occurred. When *K*_leaf_ was higher than 8.70 mmol m^−2^ s^−1^ MPa^−1^, *g*_s_ decreased slowly with *K*_leaf_ decreasing. The plant’s ability to transport water efficiently through the xylem is greatly diminished, leading to a rapid decline in *g*_s_ as a protective mechanism to prevent further water loss. This threshold effect suggests that plants have evolved specific physiological mechanisms to respond to varying degrees of soil water stress. When *K*_leaf_ remains above the threshold, plants can maintain relatively high *g*_s_ levels, balancing the need for carbon assimilation through photosynthesis with the risk of excessive water loss. However, as soil water stress intensifies and *K*_leaf_ decreases below the threshold, plants rapidly close their stomata to minimize water loss, even if it comes at the cost of reduced carbon assimilation. The reasons for this difference in stomatal behavior compared to previous studies that observed a linear correlation between *g*_s_ and *K*_leaf_ in rice [25] are likely multifaceted. One possibility is that different plant species exhibit varying degrees of sensitivity to soil water stress, with some species being more reliant on stomatal closure as a protective mechanism than others. Additionally, environmental factors such as temperature, humidity, and soil texture can influence stomatal behavior and the relationship between *g*_s_ and *K*_leaf_. 

We observed that *g*_s_ started to decrease rapidly in correlation with *Ψ*_MD_ (Figure 2b). It was less than embolism (P_50_, Figure 3). This result is similar to previous studies. Martin et al. [26], through a meta-analysis of functional traits associated with stomatal response to drought, revealed that in most species, stomatal closure precedes the occurrence of embolism during periods of water scarcity. Similarly, Chen et al. [27] reported that Ginkgo trees closed their stomata before a significant loss of xylem hydraulic conductivity. These observations support the view that stomatal behavior is a primary mechanism for plants to regulate water status and avoid water loss during drought stress.

Notably, the relationship between *g*_s_ decline and xylem embolism was not consistent across all conditions. Corso et al. [11] demonstrated that the decrease in *K*_leaf_ and *g*_s_ in wheat under soil drought was not attributed to xylem embolism but might be induced by changes in the water absorption capacity of roots. This suggests that the interaction between stomatal behavior and xylem embolism may vary among plant species and under different soil drought conditions. Our results further indicated that as soil water deficit reached severe levels (M and L treatments), the decline in *g*_s_ was primarily driven by *K*_leaf_ (Figure 2a). This suggested that under severe water stress, plants may regulate water transport in the leaves to further reduce water loss. Such regulation could be associated with the redistribution of water within the plant and its preferential allocation to vital life processes such as photosynthesis.

Understanding the interaction between stomatal closure, which regulates water loss through transpiration, and xylem embolism, which affects water transport in the plant vasculature, is key to comprehending plant adaptation strategies. In our study, we observed that as soil water deficit surpassed a certain threshold (under M and L), the decline of *g*_s_ was primarily driven by changes in *K*_leaf_ (Figure 2a). This finding highlights the intricate interplay between stomatal behavior and xylem embolism in plants experiencing water stress. Notably, no significant difference in P_50_ was observed between H and L (Figure 3). This could be explained by the fact that cavitation, the formation of air embolisms in the xylem, is a common occurrence in crops, including rice, and that rice plants have the ability to quickly reverse cavitation [9]. This resilience to embolism reversal is crucial for plants to maintain water transport efficiency and survive drought stress. Furthermore, studying the interaction between stomatal behavior and xylem embolism under different environmental conditions could provide valuable information for predicting plant responses to climate change. In summary, our study provides new insights into how plants cope with water stress through stomatal behavior and xylem embolism. Future research should further explore the complexity of this interaction across different plant species and environmental conditions, as well as how these mechanisms influence plant adaptability and survival. This will aid in more precise predictions and responses to the impacts of climate change on global plant communities.

The revelation of diverse stomatal regulation strategies exhibited by winter wheat under varying drought intensities is indeed groundbreaking. Traditionally, wheat grown in pots has been known to adopt an anisohydric stomatal regulation, maintaining a relatively high *g*_s_ even under conditions of declining *K*_leaf_ [10]. Our findings were consistent with this observation, indicating that under mild drought (H or early-stage drought), wheat stomata tend to regulate anisohydrically, with *g*_s_ not experiencing a significant drop as *K*_leaf_ decreases. This allows for a larger stomatal aperture, potentially enabling a higher influx of CO_2_ into the leaves. This response represents a radical growth strategy, crucial for plant survival and productivity during periods of moderate water stress. However, our results also demonstrated that wheat stomata tend to regulate in a more isometric strategy with an increased intensity of soil drought stress (Figure 2). This transition in stomatal regulation strategies suggests that wheat exhibits a complex and adaptive synergism between stomatal and hydraulic regulatory mechanisms.

The ability of crops to regulate water loss through the modulation of stomatal pore size has been the focus of numerous investigations [26]. The SL_en_ and SD are key stomatal traits that govern the regulation of *g*_s_ [28]. Our results indicated that SL_en_ on both the adaxial and abaxial surfaces remained unaffected by varying soil water levels (Figure 5a). This finding suggests that under the experimental conditions, SL_en_ may not be a primary regulator of *g*_s_ in response to soil drought stress. In contrast, SD displayed a distinct pattern, with significantly higher SD on the adaxial surface in L compared to H (Figure 5b). Moreover, SD on the abaxial surface increased significantly with increasing soil moisture of the three soil water levels.

*g*_s_ and SD showed a significant negative correlation (Figure 6a), which corresponds to a previous study [20], suggesting that the number of stomata per leaf area plays a crucial role in modulating *g*_s_. Under drought conditions, plants tend to increase SD, likely as a compensatory mechanism to maintain gas exchange while minimizing water loss. This phenomenon was further supported by the observation that the decrease in *g*_s_ and *V*_cmax_ under soil drought was mediated by the increase in SD (Figure 6). These findings highlight the importance of SD as a primary response mechanism to drought stress in plants. By adjusting the number of stomata, plants can optimize water use efficiency and maintain physiological functions during periods of water scarcity. In summary, our research provides insights into the role of stomatal traits, particularly SD, in regulating *g*_s_ and plant responses to drought stress. The findings have the potential to contribute to the development of drought-tolerant crop varieties and sustainable agricultural practices.

Our findings revealed that SPI in the L treatment was significantly higher than in H. This observation suggests that SPI may be a contributing factor to the difference in *g*_s_ under soil water stress. The elevated SPI in plants under drought conditions likely reflects an adaptive strategy to maximize photosynthetic capacity per unit leaf area, compensating for the reduced water availability [29]. Interestingly, our results also revealed a significant decrease in *V*_cmax_ and *J*_max_ under L compared to H (Table 1). This finding was inconsistent with previous reports that showed no significant changes in *V*_cmax_ and *J*_max_ in drought-treated trees [30]. The discrepancy in these results indicates that crops may be more sensitive than trees to drought stress, experiencing a decline in their photosynthetic potential. The combined increase in SPI and decrease in *V*_cmax_ and *J*_max_ observed in our study could be interpreted as an adaptive response of plants to soil drought, aimed at reducing energy consumption while maintaining photosynthetic efficiency. In addition, *g*_s_ was also regulated by ABA in our study, and the relationship between *g*_s_ and ABA showed a significant negative correlation (Figure 4c), consistent with the results of previous studies. For instance, Chen et al. [27] and Hussain et al. [31] found that soil water stress caused a significant increase in ABA, which was consistent with our results (Table 1). These findings further support the role of ABA in mediating *g*_s_ responses to drought stress. In conclusion, the present study provides valuable insights into the adaptive responses of plants to drought stress, particularly with regard to SPI, V_cmax_, *J*_max_, and ABA-mediated regulation of *g*_s_. These findings have important implications for developing effective water management strategies for sustainable agriculture.

## 4. Materials and Methods

### 4.1. Experimental Site and Design

The pot trials were conducted in a climate-controlled growth chamber at the Experimental Station of the China Institute of Water Resources and Hydropower Research, located in Daxing District, Beijing. Temperatures were maintained at 25 °C during the day and 18 °C at night, and the relative humidity was controlled at 75% during the day and 80% at night in the growth chamber. The photoperiod was set for 12 h, from 7:00 A.M. to 7:00 P.M., with photosynthetically active radiation surpassing 100 μmol m^−2^ s^−1^, provided by a combination of sunlight and LED lamps (Signify Holding company, Eindhoven, The Netherlands). Dynamic variations of temperature and humidity in the growth chamber are shown in Figure 7a. 

Twenty seeds of the “*Jimai 22*” (Qingfeng Seed Industry Company, Cangzhou, China) wheat variety were sown in each plastic pot, each with a capacity of 5.6 L and dimensions of 24 cm in upper diameter, 20 cm in bottom diameter, and 25 cm in height. The pots were filled with 2.5 kg of mixed soil, consisting of a 1:1 mixture of nutrient-rich soil (organic matter >90%, pH 5–6, EC 2–3 mS/cm) and local topsoil from the experimental site (0–30 cm depth). The soil texture was classified as loam according to the international standard for soil texture classification. Field capacity (FC) and bulk density of the mixed soil were 26% and 1.20 g/cm^3^. 

The wheat plants were well-watered for a week before the soil drought treatment. Each pot received 1.6 g of urea every two weeks after the drought treatment. To reduce soil surface water evaporation, we added a 2-cm layer of perlite on top of each pot. According to previous studies by Ding et al. [32] and Mu et al. [33], we set three treatments: high soil water supply (H), which was maintained at a moisture content of 85–95% FC; moderate soil water supply (M), which was maintained at a moisture content of 65–75% FC; and low soil water supply (L), which was maintained at a moisture content of 45–55% FC. Soil moisture in all pots was measured daily using the weighing method, and the soil moisture dynamics at three different soil water supply levels are shown in Figure 7b. The soil volumetric water content during the three treatment periods was 25.61%, 18.01%, and 12.03% in H, M, and L, respectively.

### 4.2. Measurements

#### 4.2.1. Photosynthetic Parameters

The *g*_s_ of the newly fully expanded flag leaf was measured using the Li-6800 portable photosynthetic system (Li-COR, Inc., Lincoln, NE, USA) between 10:00 and 13:00. The Li-6800 was equipped with a 2 cm^2^ leaf chamber. The photosynthetic photon flux density of the leaf chamber was set at 1200 μmol m^−2^ s^−1^. The temperature of the leaf chamber was set at 26 °C. The CO_2_ concentration of the leaf chamber was controlled at 400 ppm by a steel cylinder containing CO_2_. The flow rate was set at 500 μmol/s. The relative humidity was maintained at 55%. Following the *g*_s_ measurements, *A*_n_-*Ci* curves were measured using the method of rapid *A*_n_–*Ci* response (RACiR) on the same leaves after *g*_s_ measurements [34]. The temperature during measurement of the leaf chamber was maintained at 25 °C, and the light intensity of the leaf chamber was set at 2000 μmol m^−2^ s^−1^. The *V*_cmax_ and *J*_max_ were calculated using the “plantecophys” package in R version 4.1.2 (https://cran.r-project.org accessed on 1 November 2021) [35]. Each treatment was replicated five times.

#### 4.2.2. Leaf Water Potential (*Ψ*_leaf_)

The *Ψ*_PD_ was measured between 6:00 and 7:00, and the *Ψ*_MD_ was measured between 11:30 and 13:30 from flag leaves and brought to the lab within 10 min of sample collection. *Ψ*_leaf_ was measured using a pressure chamber (PMS Instrument Company, Albany, OR, USA). The leaf incision was made on the air chamber lid and then the air chamber was tightened. It was slowly pressurized, and the incision was examined with a magnifying glass. Once small droplets of water formed at the incision, the pressure application was immediately stopped, and the pressure value was recorded. Each treatment was replicated five times.

#### 4.2.3. Leaf Hydraulic Conductivity (*K*_leaf_)

Wheat plants with fully developed flag leaves were selected, and their flag leaves were excised using scissors, leaving approximately 5 cm of the sheath. The selected plants were immediately submerged in water, and a second pruning was done to eliminate any remaining air bubbles from the stem. Subsequently, the cut leaves were placed in a dark environment for rehydration, and then exposed to an irradiation lamp to induce the opening of the leaf stomata. After reaching a steady state for 30 min, the transpiration rate (*E*) was measured using the LI-COR 6800 photosynthesis measurement system (LI-COR, Inc., USA). The *E* was recorded once the leaves reached a stable transpiration state. Subsequently, the leaves were quickly placed into a sealed bag with damp paper towels. Then, they were kept in a dark environment for about 30 min to achieve equilibrium before measuring the *Ψ*_leaf_. Each treatment was replicated five times.

The *K*_leaf_ was calculated as follows [36]:*K*_leaf_ = *E*/(0 − *Ψ*_leaf_)(1)

#### 4.2.4. Leaf Vulnerability Curves

Twenty tillering stems with three fully developed leaves were chosen and cut before the lights were turned on. Subsequently, the stems were chopped and placed in water to hydrate. Initially, we measured the maximum hydraulic conductance (*K*_max_) of the middle leaf of the aforementioned stems using the method described above. Following this, the remaining wheat stems were dehydrated on the laboratory table for varying durations.

Subsequently, the stems with varying degrees of dehydration were wrapped in black plastic bags and left to stabilize for at least 30 min to measure the leaf water potential. The initial leaf water potential values (*Ψ*_0_) were calculated as the average of the top and bottom *Ψ*_leaf_ values.

The difference in leaf water potential between the top and bottom leaves was below 0.1 MPa (below 0.3 MPa for severely dehydrated). The percentage loss of hydraulic conductance (PLC) was calculated as follows [37]:PLC = 100 × (1 − *K*_leaf_/*K*_max_)(2)

The least squares method, based on empirical functions, was used to fit the vulnerability curves for H and L. This was done to determine the xylem pressure (P_50_) at 50% hydraulic conductance loss and the slope (S) of the vulnerability curve at the inflection point (P_50_) [11]:PLC = a/(1 + exp(S(*Ψ*_0_ − c))(3)

The *K*_leaf_ loss is expressed as a percentage of change per megapascal (%/MPa). The slope (S) reflects the sensitivity of embolism spread in the xylem, and a steeper S indicates a faster spread of embolism. One vulnerability curve was measured for each treatment. The vulnerability curve is modeled using a Weibull function from the “fitplc” package [38] in R 4.1.2 (https://cran.r-project.org accessed on 1 November 2021).

#### 4.2.5. Leaf Relative Water Content

The newly fully expanded flag leaves were weighed, then soaked in distilled water, weighed again, and finally weighed after drying. The relative water content (RWC) of the leaves was calculated using the following formula [39]:RWC = (W_f_ − W_d_)/(W_t_ − W_d_)(4)

W_f_ represents the fresh weight of leaves; W_t_ is the saturated weight of the leaves after they have been fully immersed in distilled water for 2 h; and W_d_ represents the weight of the leaves after they have been fixed at 105 °C for 30 min and then dried in a 70 °C oven for 72 h. Each treatment was measured in three replicates.

#### 4.2.6. Abscisic Acid Content (ABA)

The flag leaves adjacent to those that had been measured for *g*_s_ under the same treatment were promptly collected. They were then wrapped in aluminum foil, rapidly frozen in liquid nitrogen, and stored in a refrigerator at −40 °C. ABA was measured using an enzyme-linked immunosorbent assay (ELISA) [40]. The BNTY kit, produced by Beijing Beinong Tianyi Biotechnology (Beijing, China), was used in the experiment. Each treatment was measured in five replicates.

#### 4.2.7. Stomatal Characteristics

Transparent nail polish was applied to the leaf surface in a thin layer with a sample size of 5 mm × 15 mm and left to air dry for 5 min. The dry nail polish was then peeled off using clear tweezers and placed on a microscope slide [41,42]. Stomata on leaf imprints were observed with a Leica light microscope (DM2500, Leica Corp, Wetzlar, Germany) connected to a camera.

Stomatal density (SD) was measured under a 10 × 10 magnification microscope by counting the number of stomata within three randomly selected fields of view. SD was then calculated as the average value divided by the corresponding image area, resulting in the number of stomata per square millimeter. Stomatal length (SL_en_) was measured under a 10 × 40 magnification microscope using Motic Panthera software version 1.0.23 (Motic China Group Limited, Xiamen, China), and the average length was calculated for three randomly selected stomata. Stomatal Pore Index (SPI) was calculated as follows [43]: SPI = S_Len_^2^ × SD × 10^−4^(5)

This index can be interpreted as an approximate measure of the proportion of the stomatal aperture area to the leaf area. SPI is an indicator of the degree of stomatal opening in crop leaves [44]. The SD represents the stomatal density, and SL_en_ represents the stomatal length in Equation (5). Each treatment’s SL_en_, SD, and SPI were measured in five replicates.

### 4.3. Statistical Analyses

SPSS 23 software (IBM, Inc., Armonk, NY, USA) was used to conduct a one-way ANOVA on the data related to *Ψ*_leaf_, *K*_leaf_, *g*_s_, stomatal traits, physiology, and biochemistry under various treatments. The differences between the means were analyzed using the least significant difference (LSD) multiple comparison test, with a significance level of 5%. Regression analysis was performed on physiological indexes using Origin 2021. A significance threshold of *p* < 0.05 was applied, and only results with significant differences were presented. The figures and tables were created using Origin 2021 (OriginLab, Northampton, MA, USA) and Microsoft Excel 2021.

## 5. Conclusions

This study revealed the integrated effects of soil water stress on winter wheat hydraulic properties and stomatal regulation. We found that ABA and stomatal traits including SD and SPI increased, while *Ψ*_PD_ and *Ψ*_MD_, *K*_leaf_, *g*_s_, *V*_cmax_, *J*_max_, and RWC, all considerably decreased under soil water stress. Piecewise linear positive relationships between *g*_s_ and *Ψ*_MD_ and *K*_leaf_ were found in this study. These results revealed that wheat’s stomatal regulation transitioned from anisohydric to isohydric during soil drought. Specifically, during mild soil drought (65–75% FC), wheat adopted anisohydric stomatal regulation, characterized by stomata remaining open to maximize carbon assimilation. Meanwhile, *K*_leaf_ and *Ψ*_leaf_ significantly decreased. Under severe drought (45–55% FC), isohydric stomatal regulation was observed, with *g*_s_ decreasing to prevent further reduction in *K*_leaf_ and *Ψ*_leaf_, thereby avoiding hydraulic imbalance. Our study also found a negative correlation between *g*_s_ and SD, as well as a negative correlation between *V*_cmax_ and SD, showing that water relations of wheat were also influenced by stomatal traits. These findings enhance our understanding of how crops regulate their stomatal strategy to balance growth and survival and provide theoretical guidance for tailoring water management practices to correspond with crop stomatal regulation strategies under varying soil moisture.

## Figures and Tables

**Figure 1 plants-13-02263-f001:**
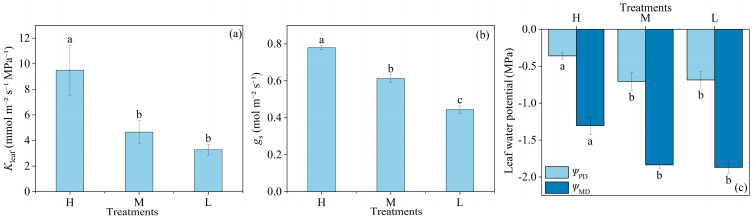
Leaf hydraulic conductivity (*K*_leaf_) (**a**), stomatal conductance (*g*_s_) (**b**), and leaf water potential at predawn (*Ψ*_PD_) and midday (*Ψ*_MD_) (**c**) under high (H), moderate (M), and low (L) soil water supply. All data represent means ± standard errors of five replicates. Different letters indicate significant differences at the *p* < 0.05 level between treatments according to Duncan’s multiple range test.

**Figure 2 plants-13-02263-f002:**
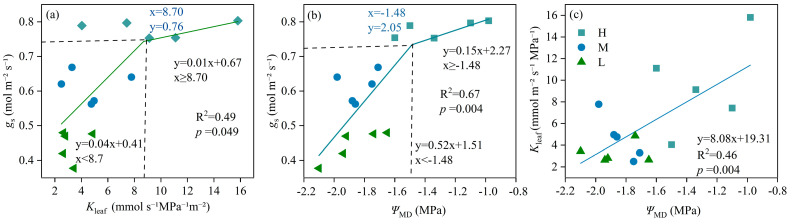
Relationships between stomatal conductance (*g*_s_) and leaf hydraulic conductivity (*K*_leaf_, (**a**)), *g*_s_ and midday leaf water potential (*Ψ*_MD_, (**b**)), and *K*_leaf_ and *Ψ*_MD_ (**c**) under high (H), moderate (M), and low (L) soil water supply. Piecewise linear regressions were used to model *g*_s_ vs. *K*_leaf_ and *g*_s_ vs. *Ψ*_MD_, and linear regression of *K*_leaf_ vs. *Ψ*_MD_ was used to fit the data. The dashed lines in (**a**,**b**) indicate the horizontal and vertical coordinates of the breakpoints, which are known as segment breakpoints. Before the point, the model shows a clear linear correlation, while, after the breakpoints, the model follows a different linear relationship.

**Figure 3 plants-13-02263-f003:**
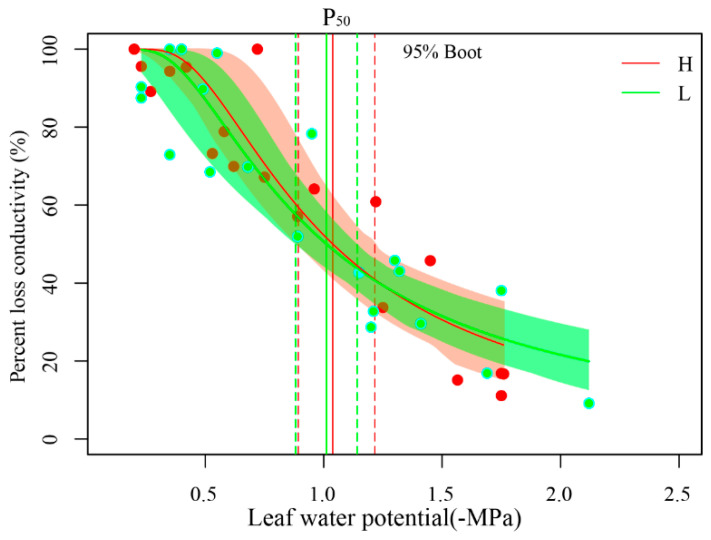
Percentage loss of wheat leaf hydraulic conductance with decreasing leaf water potential under high (H) and low (L) soil water supply treatments. The vertical solid line in each curve represents leaf water potential at 50% hydraulic conduction loss (P_50_), while the vertical dashed lines indicate its 95% confidence interval. The light green area for the low treatment (L) and the light red area for the high treatment (H) represent the standard error of the estimated parameters.

**Figure 4 plants-13-02263-f004:**
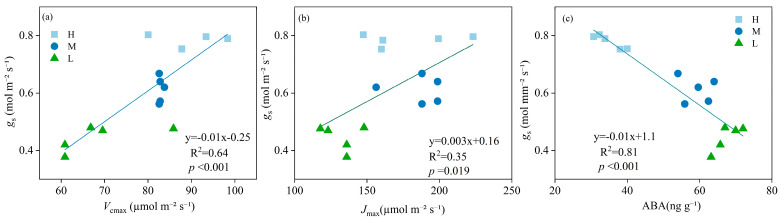
Relationships between stomatal conductance (*g*_s_) and maximum rate of carboxylation (*V*_cmax_), (**a**); *g*_s_ and maximum photosynthetic electron transport rates (*J*_max_), (**b**); *g*_s_ and abscisic acid content (ABA), (**c**) under high (H), moderate (M), and low (L) soil water supply. Linear regressions of *g*_s_ vs. *V*_cmax_, *g*_s_ vs. *J*_max_, and *g*_s_ vs. ABA were used to fit the data.

**Figure 5 plants-13-02263-f005:**
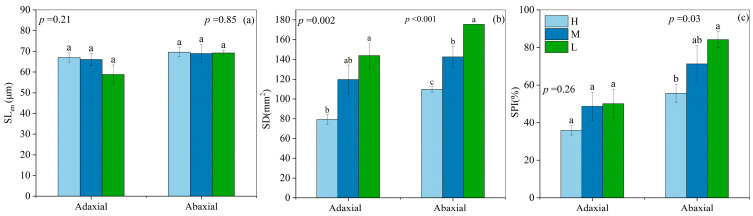
Stomatal traits on the adaxial and abaxial epidermis under high (H), moderate (M), and low (L) soil water supply, including stomatal length (SL_en_, (**a**)), stomatal density (SD, (**b**)), and stomatal pore index (SPI, (**c**)). All data represent the means ± standard errors of five replicates. Different letters indicate significant differences at the *p* < 0.05 level between soil water levels according to Duncan’s multiple range test.

**Figure 6 plants-13-02263-f006:**
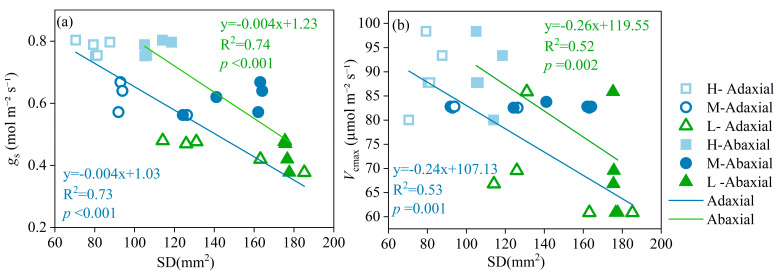
Relationships between stomatal conductance (*g*_s_) and stomatal density (SD) (**a**), and maximum rate of carboxylation (*V*_cmax_) and SD (**b**), on the adaxial and abaxial epidermis under high (H), moderate (M), and low (L) soil water supply. Linear regressions of *g*_s_ vs. The SD and *V*_cmax_ vs. SD were used to fit the data. The lines are colored green for the adaxial side and light blue for the abaxial side.

**Figure 7 plants-13-02263-f007:**
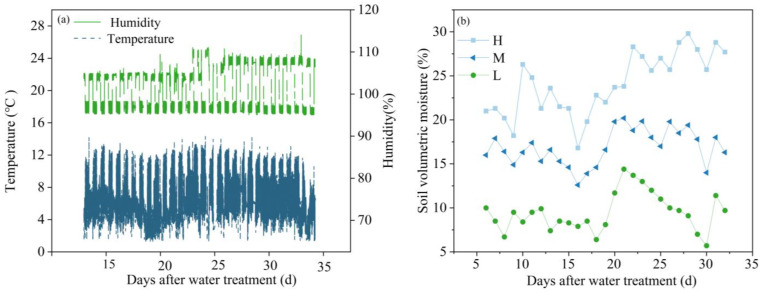
Dynamics of temperature, humidity, and soil moisture in the artificial climate chamber during the experimental treatment (**a**) and soil moisture dynamics under high (H), moderate (M), and low (L) soil water supply (**b**).

**Table 1 plants-13-02263-t001:** Biochemical indicators of different water treatments.

Treatments	*V*_cmax_ (µmol m^−2^ s^−^¹)	*J*_max_ (µmol m^−2^ s^−^¹)	RWC (%)	ABA (ng g^−1^)
H	89.87 a	182.49 a	0.61 a	34.95 c
M	83.05 a	181.05 a	0.54 a	59.19 b
L	65.73 b	131.18 b	0.37 b	66.17 a

Note: All data represent the means of five replicates. Different letters indicate significant differences at the *p* < 0.05 level between treatments according to Duncan’s multiple range test. H: high soil water supply; M: moderate soil water supply; L: low soil water supply. *V*_cmax_, maximum rate of carboxylation; *J*_max_, maximum photosynthetic electron transport; RWC, relative water content; ABA, abscisic acid.

## Data Availability

The original contributions presented in this study are included in the article. Further inquiries can be directed to the corresponding author.

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
