# Peer review of "Integrated Effects of Soil Moisture on Wheat Hydraulic Properties and Stomatal Regulation"

_plants, 2024, doi:10.3390/plants13162263_

Round 1
Reviewer 1 Report
Comments and Suggestions for Authors
The paper reports data on hydraulic and stomatic parameters in winter wheat grown in pots. The article is interesting and well-structured. However, due to the large number of parameters and variables discussed, I found it difficult to follow the text, mainly because the acronyms and their respective definitions were not used correctly. Regarding the acronyms, please adhere to the general guidelines to ensure clarity and consistency throughout the text. Here are the best practices:
1. Initial Definition: Each acronym should be defined the first time it is mentioned in the main text of the paper. This includes the introduction, where you present the acronyms and their full meanings.
2. Abstract: It is common practice to define acronyms in the abstract if the acronym is crucial for understanding the content. Defining acronyms in the abstract helps ensure that readers immediately understand what the acronyms mean, especially if the abstract is read independently of the rest of the paper.
3. Conclusions: Generally, it is not necessary to redefine acronyms in the conclusions, provided they have already been defined earlier in the text. The conclusion should assume that the reader is already familiar with the acronyms used throughout the paper.
4. Consistent Usage: After the initial definition, use the acronym consistently throughout the text. Ensure that the first definition is clear and understandable, and avoid introducing new acronyms without proper definition.
In relation to methods, please mention the soil properties (e.g., texture, density, porosity, or other parameters) and ensure that all the parameters determined and presented in the methods were included in the results, analysys and discussion.
Please see my specific comments in the pdf attached.

Reviewer 2 Report
Comments and Suggestions for Authors
Dear authors,
I made comments and suggestions on your manuscript. Please go through the manuscript and make the necessary edits.
General comments:
· Add all equipment (Manufacturing company, state, country) used in the experiment.
· Check where to use the word “respectively” throughout the manuscript.
Line 18 – 19: “Results showed... and gs was reduced by 21.20% and 43.41% under M and L conditions, respectively.
Line 23: “… Stomatal density (SD) and stomatal pore area index on the abaxial side were increased by 59.80% and 23 52.30%, respectively, under L.”
Instead … Stomatal density (SD) and stomatal pore area index under L on the abaxial side were increased by 59.80% and 23 52.30%, respectively.
Line 119 – 120: The bars in Figure 1(c) should contain one color so that leaf water potential at predawn (ΨPD) [crossed-pars] and midday (ΨMD) [open bars] can be clearly seen.
Line 133 – 136: What is the dashed line in Figure 2 (a) and (b) are indicating? Please provide a brief description of these dashed lines.
Line 157 – 158: Please spell out as a footnote all the Biochemical indicators in Table 1.
Line 356: “… combination of sunlight and LED 356 lamps (Manufacturing company, state, country).
Line 358: … Twenty seeds of the "Jimai 22" wheat variety were sown in each plastic pot (Manufacturing company, state, country).
Line 374 – 375: Figure 7a, please assign one of the variables (Temperature or humidity) to secondary Y-axis.
Line 391: Check these numbers in “…Inside the leaf chamber, CO2 concentration was set to 400, 300, 200, 150, 100, 50, 400, 600, 800, 1000, 1500, and 2000 ppm…”. There are two set of numbers: 400, 300, 200, 150, 100, 50 and 400, 600, 800, 1000, 1500, and 2000 ppm; otherwise, there are a repetitive number and lacks consistency (ascending order?).
Line 464: Stomatal density (SD) was measured under a 10x10 (unit? Micro/mm?) microscope…
Line 470 – 471: Spell out what SLen2 × SD in the Equation SPI = SLen2 × SD × 10-4?
Line 483 – 509: Conclusion: The conclusion should contain brief findings of your research can be done in one paragraph.
With regards,
The reviewer
